# OpenReview forum: "Measuring the Faithfulness of Thinking Drafts in Large Reasoning Models"
_NeurIPS.cc/2025/Conference — NeurIPS 2025 poster_

### Official Review · Reviewer_wfk8 · 2025-06-30

**Clarity:** 3
**Significance:** 3
**Originality:** 2
**Rating:** 4
**Confidence:** 3

**Summary:**

The paper proposes a systematic counterfactual intervention framework to evaluate the faithfulness of thinking drafts in Large Reasoning Models (LRMs). It introduces two dimensions of faithfulness: Intra-Draft Faithfulness, which examines whether individual reasoning steps causally influence the draft’s conclusion, and Draft-to-Answer Faithfulness, which assesses whether the final answer aligns with and depends on the thinking draft. The study conducts experiments on six state-of-the-art LRMs using datasets like GPQA and MMLU, employing counterfactual step insertions and perturbations to test these dimensions. The findings reveal selective integration of reasoning steps, with higher faithfulness to backtracking steps, and frequent misalignment between the draft’s conclusions and final answers. The paper concludes that current LRMs exhibit limited faithfulness, highlighting the need for improved interpretability and reliability in reasoning processes. While the study provides valuable insights, it relies on automated evaluation with GPT4o, lacks detailed descriptions of counterfactual step construction, and employs relatively standard methods with limited methodological innovation.

**Questions:**

Questions
1) Can the authors provide a detailed explanation of how counterfactual reasoning steps are designed and integrated into the thinking drafts? Specifically, what criteria ensure these steps are “meaningfully tied to the reasoning process” and “verifiable by an external LLM-based evaluator,” as mentioned in Section 4.2?
2) Given the exclusive use of GPT-4O-MINI for decomposing and labeling reasoning steps, have the authors considered validating these annotations against human evaluations to ensure accuracy and mitigate potential model biases?

**Ethical Concerns:**

["NO or VERY MINOR ethics concerns only"]

**Limitations:**

This paper does not discuss limitations. (The author claims that limitations are mentioned in the conclusion, but in reality, they are not.)
The limitations are listed in the questions.

**Quality:**

2

**Strengths And Weaknesses:**

Strengths
1) The evaluation includes six diverse LRMs, covering different model sizes, post-training strategies (RLVR-based and distillation-based), and task complexities (GPQA and MMLU). This broad scope strengthens the generalizability of the findings across various model architectures and datasets
2) The adoption of counterfactual step insertions to assess Intra-Draft Faithfulness is a robust method for probing causal dependencies within reasoning steps. This approach effectively tests whether the model integrates or corrects intermediate steps, providing insights into the reasoning process’s integrity.
3)  By evaluating models using thinking drafts from DeepSeek-R1, Qwen3-32B, and self-generated drafts, the study ensures a controlled and consistent evaluation, mitigating biases from a single draft source and enhancing the robustness of the results.


Weaknesses
1) The evaluation relies entirely on GPT-4O-MINI for decomposing and labeling thinking drafts, as noted in Section 4.2. This automated approach may introduce biases or inaccuracies, as it lacks validation against human judgment, which is critical for ensuring the reliability of faithfulness assessments.
2) The paper mentions inserting counterfactual reasoning steps to evaluate Intra-Draft Faithfulness but does not provide detailed methodology on how these steps are constructed (e.g., their content, structure, or selection criteria). This omission, evident in Section 4.2, limits reproducibility and transparency.
3) While the counterfactual intervention framework is systematic, it builds heavily on existing methods like CoT faithfulness evaluations and simulability metrics (Section 2). The paper does not introduce novel techniques, reducing its contribution to advancing evaluation methodologies.

---

> ### Author Rebuttal · Authors · 2025-07-31
>
> Thank you very much for your valuable feedback, constructive comments, and appreciation of our experimental setup. In response, we have added an additional examination of LLM judge reliability involving **human and cross-model annotation** (see Answer 1). Below, we carefully address each of your concerns.
>
> ---
> ### **Answer 1 (Weakness 1 & Question 2):** The LLM-as-a-judge approach may introduce biases or inaccuracies, as it lacks validation against human judgment.
> We appreciate your concern regarding the reliability of using LLMs as evaluators. To rigorously verify the reliability, we conducted additional experiments involving three human annotators and multiple judging models and measure their agreement rates (i.e., the percentage of matching annotations between two annotators) and Cohen’s Kappa coefficient:
>
> * **Step Labeling:** 3 annotators evaluated 200 randomly sampled reasoning steps (100 self-reflect, 100 continue reasoning) from MMLU Deepseek-R1 traces.
> * **Behavior Classification and Trace Answer Extraction:** 3 annotators evaluated 200 reasoning traces from Qwen3-14B (100 explicitly corrected, 100 consistently followed) across all intervention types on MMLU for Intra-Draft Faithfulness.
>
> **Human-Human Agreement Rates:**
> | Task | Aveg Agreement (%) |
> |---   | --- |
> | Step Labeling | 90.7 |
> | Behavior | 81.0 |
> | Answer Extraction | 99.3 |
>
> **Human-LLM Agreement (using Majority Vote from human annotations):**
> | Task | Agreement (%) |
> | --- | ------------- |
> | Step Labeling | 91.5 |
> | Behavior | 81.0 |
> | Answer Extraction | 100 |
>
> We also calculate **Cohen's Kappa coefficient** between the majority vote of human labels and the LLM for both step labeling and behavior classification. Step labeling yields a coefficient of 0.83, indicating **almost perfect agreement**. Behavior classification, which is more challenging for humans due to the extended content in the reasoning trace, still yields a coefficient of 0.62, reflecting **substantial agreement**.
>
> Furthermore, we assessed reliability across multiple LLM judges (Qwen2.5-32B-Instruct, Llama-3.2-70B-Instruct, Qwen3-32B) for behavior classification on three additional models: Deepseek-R1-Distill-1.5B (DS-1.5B), Skywork-OR1-7B (OR1-7B), and Qwen3-14B (faithfulness rate reported in Answer 5 to Reviewer hiiz) over all generated traces on both MMLU and GPQA:
>
> **Cross Model Agreement Rates:**
>
> | Models | Avg Agreement (%) |
> | --- | --- |
> | DS-1.5B |  91.5 |
> | OR1-7B | 81.1 |
> | Qwen3-14B | 82.4 |
>
> These high agreement rates, **comparable to human-level consistency**, strongly support the reliability of our LLM-based evaluation method.
>
> ---
>
> ### **Answer 2 (Weakness 2 & Question 1):** The detailed construction (e.g., content, structure, selection criteria) of counterfactual reasoning steps is omitted.
>
> Thank you for raising this point. We would like to respectfully clarify that these details are, in fact, provided in our paper.  Specifically:
> * **Figure 2 (Section 4.2.1) and Figure 4 (Appendix B.1)** provides clear examples illustrating the inserted counterfactual reasoning steps.
> * **Appendix B.1** explicitly present comprehensive details on selection criteria for insertion locations, the specific content and structure of inserted counterfactual steps, and the precise prompts employed to generate corrupted options.
>
> We acknowledge, however, that including these details primarily in the Appendix may limit visibility and immediate accessibility. Thus, we will emphasize these methodological specifics more explicitly in the main manuscript to enhance reproducibility and transparency with later revision.
>
> ---
>
> ### **Answer 3 (Question 1):** What criteria ensure inserted intervention steps are “meaningfully tied to the reasoning process” and “verifiable by an external LLM-based evaluator”?
>
> Thank you for this important question. To satisfy the two criteria, our interventions are carefully designed to mimic naturally occurring reasoning steps—specifically, option restatements (e.g., “Let me verify the options: A)... B)...”)—which are frequently observed in the thinking drafts of multiple-choice reasoning tasks. We manipulate the inner options to meet the criteria as follows:
>
> * **Meaningfully tied to reasoning:**
> Since multiple-choice questions inherently require explicit consideration and evaluation of provided options, our interventions directly engage critical reasoning content. By altering these option statements, we ensure that the inserted counterfactuals integrate seamlessly into the logical reasoning flow and can have global implications for the model’s reasoning trajectory.
> * **Verifiable by external evaluators:**
> Since options are central to the reasoning process, model responses to these interventions are clearly observable in the generated drafts. Models either explicitly correct the manipulated options or consistently follow (or ignore) the alterations, allowing behavior to be reliably classified into two categories: explicit correction or consistent following. This clear dichotomy enables robust external verification using carefully designed LLM-based classification prompts, as detailed in Appendix B.1.
>
> ---
>
> ### **Answer 4 (Weakness 3):** The paper does not introduce novel techniques, reducing its contribution to advancing evaluation methodologies.
> Thank you for raising this point about technical novelty. We agree that counterfactual interventions have been widely used in prior work on CoT faithfulness. However, our core contribution lies in clearly defining and operationalizing the novel concept of **thinking draft faithfulness**, and rigorously applying counterfactual interventions to systematically investigate this property in emerging LRMs.
>
> Specifically, we introduce:
>
> * A precise conceptualization of faithfulness within thinking drafts, formalized through two new perspectives: **Intra-Draft Faithfulness** and **Draft-to-Answer Faithfulness**.
> * Tailored counterfactual interventions specifically designed to address the inherent complexity and multi-path exploratory nature of reasoning drafts in LRMs. To our knowledge, **no existing interventions** in prior work effectively capture the intricate causal dependencies that emerge within the thinking drafts of recent LRMs.
>
> Thus, while our work builds upon the general idea of counterfactual intervention, the adapted definitions, novel metrics, and customized intervention and evaluation framework constitute meaningful methodological advances. These components are essential for rigorously assessing faithfulness in this emerging context. We will clarify and emphasize this contribution more explicitly in the revised manuscript.
>
> ---
> We sincerely appreciate your feedback, which significantly helps us improve the clarity, rigor, and impact of our work.

---

> > ### Author Response · Authors · 2025-08-05
> >
> > Dear Reviewer wfk8:
> >
> > Thank you again for your thoughtful and detailed review, as well as your appreciation of our paper. We sincerely value the time and effort you have devoted to evaluating our work.
> >
> > As the Author–Reviewer discussion phase nears its end, we would like to kindly ask whether our responses have addressed your concerns. We hope to have enough time to provide further clarification or conduct additional analyses if you have any remaining questions.
> >
> > In particular, to address your earlier points, we have added a reliability analysis of using LLMs as evaluators for our proposed faithfulness metrics (**Answer 1**), clarified the construction of our counterfactual reasoning steps (**Answers 2 and 3**), and explained the novelty of our proposed framework in advancing evaluation methodologies (**Answer 4**).
> >
> > Please let us know if these additions resolve your concerns or if there are any remaining issues you would like us to address.
> >
> > Best,
> >
> > Authors

---

### Official Review · Reviewer_VqHF · 2025-07-02

**Clarity:** 3
**Significance:** 2
**Originality:** 2
**Rating:** 4
**Confidence:** 3

**Summary:**

This paper introduces a systematic framework for evaluating the faithfulness of thinking drafts in Large Reasoning Models like OpenAI o1, DeepSeek R1, and Claude 3.7 Sonnet. The authors propose two key dimensions of faithfulness: (1) Intra-Draft Faithfulness - whether reasoning steps causally influence subsequent steps and conclusions within the draft, and (2) Draft-to-Answer Faithfulness - whether final answers align with and depend on the thinking draft. Through counterfactual interventions on six state-of-the-art LRMs, they find that current models demonstrate selective faithfulness to intermediate reasoning steps and frequently fail to align with draft conclusions.

**Questions:**

1. The current definition of Draft Reliance $\mathbb{1}[M(x, T', G) = M(x, T', \emptyset)]$ seems counterintuitive. Shouldn't a model be considered more draft-reliant when M(x, T', G) ≠ M(x, T', ∅), indicating that the draft influences the final decision?
2. The paper uses thinking drafts generated by Qwen3-32B and DeepSeek-R1 to evaluate smaller models, rather than evaluating these larger models directly on their own drafts. What is the specific purpose of this cross-model evaluation strategy?
3. Given that your evaluation relies on introducing artificial perturbations that fundamentally disrupt the natural reasoning flow, how can we be confident that the observed faithfulness patterns reflect genuine model behaviors rather than artifacts of the intervention process? Have you considered comparing these results with faithfulness assessments under natural, unperturbed conditions to validate the consistency of your findings?

**Ethical Concerns:**

["NO or VERY MINOR ethics concerns only"]

**Final Justification:**

After considering the authors' comprehensive rebuttal and additional experiments, I have raised my score from 3 to 4. The authors have adequately addressed several key concerns: (1) they expanded the evaluation to include MMLU College Mathematics and provided human annotation validation for LLM judge reliability, (2) they clarified the relationship between their Draft-Answer Consistency metric and simulatability-based approaches through additional comparative analysis, and (3) they provided reasonable justifications for their cross-model evaluation strategy. However, limitations remain unresolved: the fundamental constraint of multiple-choice questions limits the ecological validity of findings for real-world open-ended reasoning tasks, and the artificial nature of counterfactual interventions may not fully capture natural reasoning dynamics. While I appreciate the authors' acknowledgment of these limitations and their suggestions for future improvements, these constraints substantially limit the generalizability of the findings. The paper makes a valuable contribution to understanding LRM faithfulness through its systematic framework and comprehensive empirical analysis across six models, but the methodological limitations prevent me from recommending stronger acceptance. I encourage the authors to more prominently discuss these limitations in the revised manuscript and elaborate on potential paths toward more naturalistic evaluation methods.

**Limitations:**

No limitations in main paper or appendix.

**Quality:**

3

**Strengths And Weaknesses:**

## Strength
1. The paper addresses a critical issue as LRMs become increasingly prevalent.
2. Tests six different LRMs from multiple families (distilled vs RLVR-tuned).
3. Analyzes various factors: model size, post-training methods, task complexity, step position.

## Weaknesses
1. The evaluation relies heavily on multiple-choice questions, from only two datasets (198 GPQA and 88 MMLU samples), which may not adequately represent the complexity and diversity of real-world reasoning tasks.
2. The framework heavily relies on GPT-4o-mini for generating corrupted options and misleading reasoning, and Qwen2.5-Instruct for answer extraction and behavioral classification. However, the paper lacks systematic analysis of these components' reliability and potential biases.
3. The authors explicitly state (Line 166) that they "care less about factual correctness and more about behavioral consistency." This perspective is problematic, as maintaining consistency on incorrect reasoning is arguably counterproductive. The paper fails to analyze the critical trade-offs between faithfulness and performance, missing opportunities to understand when unfaithfulness might actually improve model outputs.
4. The evaluation approach fundamentally alters the reasoning context through external interventions, forcing models to respond to artificially corrupted states.

---

> ### Author Rebuttal · Authors · 2025-07-31
>
> Thank you for your insightful and detailed feedback. To address your questions, we have added the following evaluations:
> * An expanded benchmark that includes a **mathematical reasoning dataset** (MMLU College Mathematics) (Answer 2).
> * An additional examination of LLM judge reliability involving **human and cross-model annotation** (Answer 3).
> * A new evaluation comparing our approach with **natural, unperturbed evaluation** (Answer 6).
>
> We address each of your concerns below:
>
> ---
> ### **Answer 1 (Weakness 1):** Evaluation relies heavily on multiple-choice questions.
> We acknowledge that multiple-choice questions may not fully capture the breadth of real-world reasoning tasks. However, this choice results from the **intrinsic difficulty** of faithfulness evaluation. Open-ended or free-form reasoning tasks introduce a **high output space** of final answers, making faithfulness analysis (especially via counterfactual interventions) infeasible at scale. Prior works similarly restrict evaluations to multiple-choice settings due to this constraint [1,2]. Furthermore, the complexity introduced within the thinking drafts of LRMs exacerbates this challenge, making multiple-choice settings a practical starting point. This design choice is also shared by recent concurrent work [1].
>
> ---
> ### **Answer 2 (Weakness 1):** Evaluation is limited to GPQA (198 questions) and MMLU (88 questions).
> We appreciate your concern about dataset diversity. While our evaluation indeed leverages two datasets, the overall scale of our analysis is broader. Specifically, our experiments cover 24 intervention variants for Intra-Draft Faithfulness and 8 variants for Draft-to-Answer Faithfulness, totaling over **9,000 tested traces**.
>
> To further broaden task coverage, we have now included evaluations on the **MMLU College Math subset** (100 college-level math questions). The results across existing models and additional models, including Qwen3-14B, Skywork-OR1-7B (OR1-7B), DeepSeek-R1-Distilled-1.5B (DS1.5B), are presented in **Answer 5** of our rebuttal to **reviewer hiiz**.
>
> ---
> ### **Answer 3 (Weakness 2):** Reliability of using LLMs for modification and judgment.
> Thank you for this important concern. We conducted a detailed reliability analysis, which is discussed in **Answer 1** of our rebuttal to **reviewer hiiz**. We include human annotations and Human-LLM agreement analysis to demonstrate the consistency and trustworthiness of our automated evaluation pipeline.
>
> ---
> ### **Answer 4 (Weakness 3):** Prioritizing Draft-Answer consistency over factual correctness is problematic. The paper fails to analyze critical trade-offs between faithfulness and performance.
> Thank you for raising this point. We respectfully disagree that our framing is problematic. As noted in Lines 164–165 of the manuscript, our primary goal is to assess whether **model decisions are causally grounded in their thinking drafts**. This property is critical for reliable monitoring and safety, aligning with the emphasis in recent works [1, 4, 5].
>
> For testing purposes, we deliberately require consistency with incorrect traces, as this reveals whether models faithfully follow textual reasoning drafts under extreme conditions. However, in practice, faithfulness should be **orthogonal** to performance. An unfaithful model may select incorrect internal reasoning traces while ignoring correct final draft answers, making actual failures less tractable and hindering oversight capabilities. For instance, Skywork-OR1-32B-Preview yields higher performance on their reported reasoning benchmarks (including AIME 24, 25, and Livecodebench) compared with QwQ-32B, while still presenting better Draft-Answer consistency. Thus, we did not explore whether unfaithful behavior can improve accuracy, as we observed no such tendency, and it is outside the scope of this paper.
>
> ---
> ### **Answer 5 (Weakness 4):**  Concern about artificial perturbations disrupting reasoning flow.
>
> We agree that counterfactual interventions may not fully reflect natural reasoning processes. However, since our goal is to analyze **causal dependencies** within the thinking draft, such interventions are **necessary and inevitable**, similar to prior work on CoT faithfulness [2] that also introduces artificial counterfactual changes.
>
> Nevertheless, we have designed interventions carefully to minimize unnatural disruptions. For Intra-Draft Faithfulness, we:
> * **Identify natural reasoning shift points** within the reasoning flow (Appendix B.1), ensuring insertions do not significantly interrupt the continuity of forward reasoning.
> * Restrict perturbations primarily to the option level, preserving the reasoning about the question itself
>
> For Draft-to-Answer Faithfulness, particularly in our "Plausible Alternation" condition, we generate **fluent and coherent alternative conclusions** using separate LLMs, thus realistically mimicking natural reasoning uncertainty.
>
> While we understand this construction may not be perfect given the complexity of LRM reasoning processes, we adopt this design to make initial steps toward measuring these properties. Future work can indeed improve the naturalness of interventions.
>
> ---
> ### **Answer 6 (Question 3):** Comparison with faithfulness assessments under natural, unperturbed conditions.
> We appreciate this suggestion. To our knowledge, existing faithfulness evaluations of LRMs under natural, unperturbed conditions primarily focus on simulatability [1]. We discuss the differences between our work and these **concurrent works** in **Lines 39–46 (Section 1)**. While both aim to assess LRM faithfulness, our goals differ. Their approach measures the likelihood of shifting decisions toward hint answers in prompts and verbalize the hint within the trace, but this does not assess whether reasoning steps are faithful to the draft answer or whether the final answer faithfully depends on the draft.
>
> To clarify the difference between the simulatability-based evaluations and our proposed method, we provide additional results using the setup in [1]. As their evaluation data are not publicly available, we reconstruct their setup using four hint types. The final faithfulness scores for nine models are as follows:
>
> **Faithfulness Score(%) [1]:**
> | Models  | GPQA | MMLU Global Facts |
> |---|---|---|
> |  DS-1p5b |53.6 | 49.0|
> |  OR1-7B |67.6 | 68.4 |
> |  DS-7B |57.7   |  74.7 |
> |  DS-8B|51.3 | 56.7 |
> |  DS-14B|50.6  |  56.7 |
> |  Qwen3-14B| 53.6  |  61.4 |
> |  DS-32B| 44.4 | 58.3 |
> |  QwQ-32B| 43.9 | 64.4|
> |  OR1-32B | 48.4 | 72.5 |
>
> We observe that simulatability-based faithfulness scores are less correlated with model size, and RLVR-tuned models achieve higher scores, consistent with findings in [1].
>
> However, this contrasts with our Draft-Answer Consistency results, where RLVR tuning leads to stronger internal preferences and reduced reliance on the draft logic. An example of such unfaithful reasoning is shown in Figure 1 (Section 3).
>
> We hypothesize that this discrepancy arises because simulatability primarily captures the *verbalization* aspect of reasoning, whereas our Draft-Answer Consistency metric offers a more **fine-grained view** of the draft’s logical dependency with the final answer.
>
> Together, our findings complement simulatability evaluations and provide deeper insight into LRM reasoning.
>
> ---
> ### **Answer 7 (Question 1):** The current definition of Draft Reliance $\mathbb{1}[M(x, T',G) = M(x, T', \emptyset)]$ seems counterintuitive.
> We apologize for any confusion caused. Our definition tests whether the final answer $y$ is determined solely by the thinking draft $T'$. Here, $G$ refers to the **additional explanation** generated during the **answer stage**. If the answer remains unchanged regardless of $G$'s presence or absence, this indicates that the final decision relies exclusively on the draft itself, implying draft reliance.
> An illustrative example comparing conditions with and without $G$ is presented clearly in Table 2 (Section 4.3.2) of the manuscript.
>
> ---
> ### **Answer 8 (Question 2):** Why evaluate small models on drafts from larger models? Why not evaluate Qwen3-32B and DeepSeek-R1 directly?
>
> As clarified in Lines 65–70 (Section 1), we evaluate each model using **both self-generated and externally generated drafts** from Qwen3-32B and DeepSeek-R1. This cross-model setup ensures **controlled and consistent comparisons**, as relying solely on self-generated drafts can introduce biases—particularly when evaluating counterfactual interventions. Indeed, we observed that Draft-Answer consistency varies between self- and externally generated drafts, reinforcing the need for controlled baselines. This is one reason we excluded Qwen3-32B and DeepSeek-R1 from direct evaluation.
>
> Another reason is the challenge of ensuring fair experimental control. We deliberately design controlled experiments to isolate different factors, such as differences in base model family, sizes, and post-training datasets and methods. While these larger models are powerful, the presence of **multiple confounding variables** would hinder fair and accurate comparisons, making it difficult to draw consistent insights. Therefore, we prioritized models with well-defined training conditions to enable rigorous and interpretable analyses.
>
> ---
>
> We thank you again for your insightful comments and hope that our responses and additional results address your concerns.
>
> [1] Chen et al. Reasoning Models Don't Always Say What They Think
>
> [2] Lanham et al. Measuring Faithfulness in Chain-of-Thought Reasoning
>
> [3] Turpin et al. Language Models Don't Always Say What They Think: Unfaithful Explanations in Chain-of-Thought Prompting
>
> [4] Korbak et al. Chain of Thought Monitorability: A New and Fragile Opportunity for AI Safety
>
> [5] Baker et al. Monitoring Reasoning Models for Misbehavior and the Risks of Promoting Obfuscation
>
> [6] Wu et al. Effectively Controlling Reasoning Models through Thinking Intervention

---

> > ### Author Response · Authors · 2025-08-05
> >
> > Dear Reviewer VqHF:
> >
> > Thank you again for your thoughtful and detailed review of our paper. We sincerely appreciate the time and effort you have devoted to evaluating our work.
> >
> > As the Author–Reviewer discussion phase nears its end, we would like to kindly ask whether our responses have addressed your concerns. While we appreciate your acknowledgment of the rebuttal, it remains unclear whether our responses have resolved your concerns. We hope to have sufficient time to provide further clarification or conduct additional analyses if you have any remaining questions.
> >
> > In particular, to address your earlier points, we have added several new experiments and results:
> >
> > * Construction of an additional evaluation subset and inclusion of three more models (**Answer 2**);
> > * A reliability analysis of using LLMs as evaluators for our proposed faithfulness metrics (**Answer 3**);
> > * Additional comparisons to **concurrent** faithfulness evaluation approaches, along with a discussion of conceptual differences (**Answer 6**).
> >
> > Please let us know if these additions resolve your concerns or if there are any remaining issues you would like us to address.
> >
> > Best,
> >
> > Authors

---

> > > ### Comment · Reviewer_VqHF · 2025-08-06
> > >
> > > Thank you for your detailed and thoughtful rebuttal. I appreciate the additional experiments and clarifications you've provided, particularly the expanded evaluation on MMLU College Mathematics and the human annotation analysis.
> > >
> > > While I acknowledge your explanation regarding the use of multiple-choice questions (Answer 1), I still believe that the limitation to multiple-choice formats, rather than open-ended questions, remains a significant constraint of this evaluation framework. Many real-world reasoning scenarios do not naturally present themselves with predefined options, and the faithfulness patterns observed in constrained settings may not generalize to more naturalistic reasoning tasks. I understand the practical challenges you've outlined, but this limitation should be more prominently discussed in the paper.
> > >
> > > Additionally, I'm particularly interested in your mention of improving intervention naturalness in future work (Answer 5). Could you elaborate on specific potential directions? For instance, would it be possible to leverage the model's own uncertainty to generate more natural counterfactuals? Perhaps sampling from the model's distribution at decision points rather than injecting externally generated corruptions? Or using the model's own alternative reasoning paths when it exhibits uncertainty? Such approaches might better preserve the ecological validity of the evaluation while still allowing for causal analysis.
> > >
> > > In recognition of your comprehensive responses and the additional experiments you've conducted, I have raised my score. However, I strongly encourage you to incorporate these discussions—particularly about the limitations of multiple-choice evaluation and potential paths toward more naturalistic interventions—into the revised manuscript. These additions would strengthen the paper by providing readers with a clearer understanding of both the current limitations and future research directions.

---

> > > > ### Author Response · Authors · 2025-08-06
> > > >
> > > > We appreciate your detailed and thoughtful feedback, and we're pleased that the additional experiments and clarifications addressed some of your concerns! We will incorporate all our discussions and additional results into the next revision.
> > > >
> > > > Regarding the limitation of evaluating with multiple-choice questions, we fully agree with your point. Indeed, real-world reasoning scenarios are diverse and frequently do not align with predefined answer formats like multiple-choice. We will discuss this constraint more explicitly in the limitations section of the revised manuscript.
> > > >
> > > > Concerning improving intervention naturalness for future work, your suggestions on leveraging a model’s own uncertainty to generate more natural counterfactuals are particularly insightful. We agree that methods such as sampling from the model’s output distribution at decision points, or utilizing alternative reasoning paths when the model exhibits uncertainty, could enhance the ecological validity of interventions. However, this approach also introduces challenges. For instance, it is difficult to guarantee that sampled interventions are capable of maintaining **global dependencies** across the entire thinking draft. Furthermore, interventions based on one model’s uncertainty **may not transfer well** to others, complicating cross-model benchmarking. These concerns are tightly connected to our definition of Intra-Draft Faithfulness.
> > > >
> > > > A potential direction we are considering involves first designing globally dependent interventions—using an external model or manual construction—that better reflect natural reasoning trajectories, and then directly optimizing the intervention prompts, similar to approaches in adversarial attacks, to minimize their perplexity while preserving their intended effect. Such prompts could be jointly optimized across multiple models to improve generalizability. We will discuss this further in our later revision.
> > > >
> > > > Thank you again for your valuable comments, which have significantly contributed to improving our work's clarity and depth.

---

### Official Review · Reviewer_VpmL · 2025-07-02

**Clarity:** 3
**Significance:** 4
**Originality:** 3
**Rating:** 5
**Confidence:** 4

**Summary:**

This paper investigates "thinking draft faithfulness" in Large Reasoning Models (LRMs), an area of study not yet explored in depth. The authors propose an evaluation framework with two components. The first, "Intra-Draft Faithfulness," is a binary metric (yes/no) that assesses if a model's reasoning steps influence its conclusion. The second, "Draft-to-Answer Faithfulness," measures the dependency of the final answer on the thinking draft. This second component contains two sub-metrics: "Draft Reliance," which identifies if the answer stage uses reasoning steps not present in the draft, and "Draft-Answer Consistency," which quantifies alignment between the draft's conclusion and the model's final answer.

Using counterfactual interventions, the authors conduct experiments on six LRMs, including models from DeepSeek, Qwen, and Skywork. The results indicate that models show more faithfulness to backward reasoning steps over forward-progressing ones. The analysis also shows that final answer generation can involve reasoning not present in the initial draft. The authors attribute the observation about backward reasoning to the introspective nature of that process.

**Questions:**

1. The reliance on GPT-4o-mini as an evaluator is my concern. How do your reported faithfulness scores change with the choice of this LLM judge? Have you explored using other LLMs, or human evaluation for a subset of cases, to validate the classification?
2. The interventions, such as incorrect mappings, are constructed. Have you considered exploring perturbations that could arise during LRM reasoning, for example, by altering factual premises or logical connections?
3. The distinction between "CORRECTION" and "FOLLOW" behaviors seems open to interpretation, in particular without a formal definition of the ϕ function. Could you for instance, using multiple models, provide inter-annotator agreement scores for this classification to demonstrate its consistency?
4. Given the complexity of reasoning, a metric with a binary output might not capture all detail. Have you considered measures of faithfulness with more gradations that could capture nuances in how models integrate or deviate from a line of reasoning?
5. How do you manage cases where a "correct" faithful behavior is not defined? What are your heuristics for determining success or failure in such scenarios?

**Ethical Concerns:**

["NO or VERY MINOR ethics concerns only"]

**Final Justification:**

Technically solid paper, I got good answers. I recommend accept, as before.

**Limitations:**

Yes

**Quality:**

3

**Strengths And Weaknesses:**

### Strengths
- The paper addresses the faithfulness of LRM reasoning traces. The framework, with its two dimensions of faithfulness, offers an interesting method for analysis. The definition of multiple metrics, rather than a single one, is a strength. The use of counterfactual interventions to probe LRM behaviors is valuable to the field.
- The approach appears to be a departure from work that came before, such as Matton et al. (2025), which uses a different set of concepts like causal analysis and KL divergence. The finding regarding backtracking faithfulness is an interesting contribution.
- The experimental design uses six models on two datasets, with interventions designed to test for global dependency.
- The structure of the paper and the articulation of its motivation and experimental setup facilitate understanding. The use of graphics, such as the example in Figure 1, helps to illustrate the concepts of faithfulness and faithlessness.

### Weaknesses
- The methodology of counterfactual intervention for explanation faithfulness exists in the literature. The contribution of this paper is the application of this method to "thinking drafts" for a class of LRMs. However, aspects of the methodology affect the paper's impact. The use of GPT-4o-mini for behavioral classification introduces a source of bias that is not quantified. The interventions, which use synthetic mappings defined as incorrect for the experiment, may not represent reasoning failures that occur in practice. The faithfulness metrics, which produce a yes/no output, may not capture the full range of LRM behavior. A more extensive discussion of these limitations would improve the paper.
- Some definitions are absent or underspecified. The function ϕ, for transforming answers under intervention, is mentioned but not defined, though Section 4.2 is cited as containing the definition. The ANS function for extracting conclusions also lacks a specification. The classification of behavior (CORRECTION vs. FOLLOW) seems to depend on interpretation; without inter-annotator agreement data, its reliability is difficult to assess.

---

> ### Author Rebuttal · Authors · 2025-07-31
>
> Thank you for your insightful and valuable comments, as well as your appreciation of our work. In response, we have added an additional analysis of LLM judge reliability involving **human and cross-model annotations** (see Answer 1).
>
> We address your specific concerns below:
>
> ---
> ### **Answer 1 (Weakness 1&2 & Question 1&3):** The use of LLM-as-a-judge introduces a source of bias that is not quantified.
> We appreciate your concern regarding the reliability of using LLMs as evaluators. To rigorously verify the reliability, we conducted additional experiments involving three human annotators and multiple judging models and measure their agreement rates (i.e., the percentage of matching annotations between two annotators) and Cohen’s Kappa coefficient:
>
> * **Step Labeling:** 3 annotators evaluated 200 randomly sampled reasoning steps (100 self-reflect, 100 continue reasoning) from MMLU Deepseek-R1 traces.
> * **Behavior Classification and Trace Answer Extraction:** 3 annotators evaluated 200 reasoning traces from Qwen3-14B (100 explicitly corrected, 100 consistently followed) across all intervention types on MMLU for Intra-Draft Faithfulness.
>
> **Human-Human Agreement Rates:**
> | Task | Aveg Agreement (%) |
> |---   | --- |
> | Step Labeling | 90.7 |
> | Behavior | 81.0 |
> | Answer Extraction | 99.3 |
>
> **Human-LLM Agreement (using Majority Vote from human annotations):**
> | Task | Agreement (%) |
> | --- | ------------- |
> | Step Labeling | 91.5 |
> | Behavior | 81.0 |
> | Answer Extraction | 100 |
>
> We also calculate **Cohen's Kappa coefficient** between the majority vote of human labels and the LLM for both step labeling and behavior classification. Step labeling yields a coefficient of 0.83, indicating **almost perfect agreement**. Behavior classification, which is more challenging for humans due to the extended content in the reasoning trace, still yields a coefficient of 0.62, reflecting **substantial agreement**.
>
> Furthermore, we assessed reliability across multiple LLM judges (Qwen2.5-32B-Instruct, Llama-3.2-70B-Instruct, Qwen3-32B) for behavior classification on three additional models: Deepseek-R1-Distill-1.5B (DS-1.5B), Skywork-OR1-7B (OR1-7B), and Qwen3-14B (faithfulness rate reported in Answer 5 to Reviewer hiiz) over all generated traces on both MMLU and GPQA:
>
> **Cross Model Agreement Rates:**
>
> | Models | Avg Agreement (%) |
> | --- | --- |
> | DS-1.5B |  91.5 |
> | OR1-7B | 81.1 |
> | Qwen3-14B | 82.4 |
>
> These high agreement rates, **comparable to human-level consistency**, strongly support the reliability of our LLM-based evaluation method.
>
> ---
>
> ### **Answer 2 (Weakness 1 & Question 2):** The designed interventions may not represent reasoning failures that occur in practice. Have you considered exploring perturbations that could arise during LRM reasoning?
>
> Thank you for raising this insightful point! We acknowledge that our counterfactual interventions may not fully capture realistic reasoning failures occurring naturally in practice.
>
> However, as discussed in Lines 138–142 (Section 3), exploring counterfactual effects within thinking drafts poses **intrinsic challenges** due to LRMs' complex multi-path exploratory nature. Our chosen perturbations—corrupting or shifting mappings of multiple-choice options—were specifically designed to enable **global dependencies** throughout the whole reasoning process, helping us to consistently evaluate faithfulness.
>
>
> We acknowledge that modifying factual premises or logical connections would better reflect natural reasoning errors. However, generating such interventions at scale and precisely tracking their subtle counterfactual effects poses significant methodological hurdles. Furthermore, this approach is unlikely to establish global dependencies throughout the reasoning process. These considerations guided the current design of our intervention as an initial exploration of the thinking draft faithfulness problem. Nevertheless, we agree that this is a crucial direction for future research, and we will explicitly address these limitations and potential extensions in our revised manuscript.
>
> ---
>
> ### **Answer 3 (Weakness 1 & Question 4):** The binary faithfulness metrics may miss nuances in LRM behavior. Have you considered measures of faithfulness with more gradations that could capture nuances in how models integrate or deviate from a line of reasoning?
>
> Thank you for raising this thoughtful concern. Indeed, the inherent complexity of LRM behavior can present diverse forms of integration or deviation in response to our interventions. However, our primary research goal is to rigorously evaluate the faithfulness property, defined as whether the final reasoning or conclusion integrates explicit reasoning steps within the thinking draft, or deviates from the reasoning steps with clear correction.
>
> We intentionally chose binary metrics to clearly delineate such core property. For example:
> * If the model **explicitly rejects the intervention (at any point)**, we classify this as **CORRECTION**, implying the reasoning draft’s global trajectory should remain unchanged.
> * Conversely, if a model integrates the new mapping or corrupted option **without explicitly rejecting it**, we classify this as **FOLLOW** behavior. Under this scenario, faithfulness implies the final conclusion should change accordingly.
>
> The reasons above lead the classified behaviors to exhibit signs of faithfulness in a straightforward and easily detectable manner. The detailed classification prompts are also provided in  Appendix B.1.
>
> Introducing additional nuance, such as degrees of integration or rejection, could potentially introduce ambiguity, making LLM or even human classification more subjective and less reliable, and may not affect too much understanding of faithfulness property alone.
>
> While we agree that more fine-grained behavioral analyses could be insightful for broader behavioral understanding of models, they are less directly relevant to our current, focused investigation of faithfulness. Nonetheless, we acknowledge that this remains an interesting avenue for further exploration and will highlight this point as future work.
>
> ---
>
> ### **Answer 4 (Weakness 2 & Question 3):** The function $\phi$ and ANS are underspecified
>
> We apologize for the confusion. Since the definition of **faithful FOLLOW** (where $\phi$ is used) differs across the two intervention settings—Shift Mapping (with target projection) and Corrupt Option (with untargeted projection)—we opted to present a simplified and unified view in Section 3 to maintain readability. The detailed specification of how faithful FOLLOW is defined under each intervention is provided in Lines 207–209 of Section 4.2.1 in text.
>
> For clarity,  we also provide a formal definition of faithful FOLLOW for both settings, using a shared option set  $\mathcal{Y} = \\{y_0 = A, y_1 = B, y_2 = C, y_3 = D\\}$:
> * **Shift Mapping.** Let
> $\phi(y\_i) = y\_{(i+1)\ \bmod 4}, \\text{for } 0 \\le i \\le 3.$ In this case, the faithful following metric is
> $$\\mathbb{1}\\bigl[\\text{ANS}(T'\_{>j+1}) = \\phi(\\text{ANS}(T))\\bigr].$$
> * **Corrupt Option.** Suppose the intervention corrupts option $y_i, 0 \le i \le 3$. The model is expected to not return the corrupted option, and the faithfulness metric is measured as
> $$\mathbb{1}\bigl[\text{ANS}(T'_{>j+1}) \ne y_i\bigr].$$
>
> Additionally, **ANS** refers to a function that extracts the final answer from a reasoning trace:
> $$\text{ANS}: \mathcal{V}^* \to \mathcal{Y},$$
> where $\mathcal{V}$ is the vocabulary space. This function is defined at Line 117 in Section 3.
> We will make these definitions more explicit and accessible in the revised version.
>
> ---
>
> ### **Answer 5 (Question 5):** How do you handle cases where "correct" faithful behavior is undefined?
> Thank you for this important question. By design, we specifically aim to minimize scenarios where faithful behavior is undefined or ambiguous. Our methodology achieves this by:
>
> * Selecting interventions targeting multiple-choice options, thus ensuring that model responses to interventions are readily detectable and interpretable within thinking drafts.
> * Implementing clear classification rules (as detailed in Answer 3).
>
> In rare instances such as repetitive gibberish output, we **explicitly exclude** these from evaluation to maintain assessment reliability and interpretability. This careful filtering further ensures we only evaluate clearly defined scenarios, thus strengthening the validity of our measured faithfulness metrics.
>
> ---
> Thank you again for your detailed and constructive feedback, which has substantially helped us clarify and improve our manuscript.

---

> ### Comment · Reviewer_VpmL · 2025-08-05
>
> Thank you for your responses. I’ve kept my rating as accept and have no further questions.

---

> > ### Author Response · Authors · 2025-08-05
> >
> > We sincerely thank you again for your thoughtful and thorough comments! We will incorporate all the discussed points into the next revision. We truly appreciate your help in improving our work!

---

### Official Review · Reviewer_hiiz · 2025-07-03

**Clarity:** 2
**Significance:** 3
**Originality:** 3
**Rating:** 4
**Confidence:** 3

**Summary:**

This paper introduces a framework to evaluate whether the intermediate reasoning steps, known as "thinking drafts," produced by Large Reasoning Models (LRMs) are a faithful representation of their problem-solving process. The authors propose a counterfactual intervention method to test faithfulness along two dimensions:
- Intra-Draft Faithfulness: This assesses if individual reasoning steps causally affect the subsequent steps and the conclusion within the draft itself.
- Draft-to-Answer Faithfulness: This evaluates if the model's final answer is logically consistent with and dependent on the thinking draft.

Based on experiments with six different LRMs, the study concludes that current models are only selectively faithful. The findings show that models are more influenced by backtracking (revision) steps than continuous reasoning. Furthermore, the final answer often does not align with the draft's conclusion because the model performs additional, independent reasoning during the final answering stage.

**Questions:**

1. Incorporate Internal State Analysis: To complement the current behavioral study, we suggest analyzing the model's internal states, such as output probabilities or logits. This could provide deeper insights into model confidence during reasoning and is a common technique in related work on hallucination detection.

2. Propose Methods for Improving Faithfulness: The paper's insightful analysis provides a strong foundation for proposing solutions. The authors could explore concrete methods to improve faithfulness, such as new training strategies or prompting techniques based on the weaknesses identified in the study.

3. Strengthen the Evaluation Benchmark: If the focus remains on evaluation, the framework's robustness could be enhanced. We recommend expanding to more diverse datasets (e.g., AIME for mathematical reasoning), validating the reliability of the LLM-based evaluators, and including statistical significance testing to better support the conclusions.

**Ethical Concerns:**

["NO or VERY MINOR ethics concerns only"]

**Final Justification:**

After carefully considering the authors' rebuttal and additional experiments, I recommend raising the score due to significant improvements in evaluation reliability and benchmark coverage. The new human and cross-model annotations (Answer 1) effectively address concerns about LLM judge biases, while the inclusion of perplexity (Answer 3) provides a valuable supplementary metric that strengthens result validation. Furthermore, the expanded benchmark with MMLU college math and additional models (Answer 5) demonstrates better generalization across more challenging tasks and diverse model comparisons. Although certain aspects - particularly methods for fundamentally improving faithfulness - remain areas for future work, these substantial enhancements directly address the core weaknesses that initially affected the evaluation. Collectively, these improvements provide more robust evidence supporting the paper's claims, warranting an upward revision of my recommendation.

**Limitations:**

yes

**Quality:**

3

**Strengths And Weaknesses:**

Strengths:
1. This paper studies the faithfulness of reasoning models, which is a very important and significant problem. As AI models become more complex, ensuring that their stated reasoning is reliable is crucial for safety and trust.
2. The paper performs a large number of experiments to test its ideas. It evaluates six different models on two datasets with different levels of difficulty. This thorough testing makes the findings stronger and more credible.
3.The paper is well-written and organized in a way that is easy to follow. The authors use figures effectively to help explain complex ideas, such as the different types of faithfulness shown in Figure 1.

Weaknesses

1. Reliance on Other LLM for Judgment: The study uses other models, like GPT-4o-MINI, to create tests and classify the results. This method might not be reliable because the "judge" LLM can also make mistakes or be biased. The paper does not discuss how trustworthy this automated evaluation is.
2. Confusing Definitions: The paper's definition of "faithfulness" is complex. It is broken down into multiple parts, such as "Intra-Draft Faithfulness" and then "Draft-to-Answer Faithfulness," which is further split into "Draft Reliance" and "Draft-Answer Consistency". This can be confusing to follow. The paper could better explain why all these different parts are needed.
3. Limited Testing Method: This research only tests models by changing text prompts and observing the text output. It does not consider other ways to evaluate the models, such as looking at their internal calculations (like logits or perplexity). Including these other methods could have provided a more complete and convincing analysis.

---

> ### Author Rebuttal · Authors · 2025-07-31
>
> We appreciate your insightful and constructive feedback. In order to address your questions, we have added the following evaluations:
> * An additional examination of LLM judge reliability involving **human and cross-model annotations** (Answer 1).
> * An additional evaluation method using **internal model calculations** (perplexity) (Answer 3).
> * An expanded benchmark that includes a **mathematical reasoning dataset** (MMLU college math) and **three additional models** (Answer 5).
>
> We further address each of your concerns in detail as follows:
>
> ---
> ### **Answer 1 (Weakness 1 & Question 3):** Reliability of using LLMs for judgment and modification.
>
>
> We appreciate your concern regarding the reliability of using LLMs as evaluators. To rigorously verify the reliability, we conducted additional experiments involving three human annotators and multiple judging models and measure their agreement rates (i.e., the percentage of matching annotations between two annotators) and Cohen’s Kappa coefficient:
>
> * **Step Labeling:** 3 annotators evaluated 200 randomly sampled reasoning steps (100 self-reflect, 100 continue reasoning) from MMLU Deepseek-R1 traces.
> * **Behavior Classification and Trace Answer Extraction:** 3 annotators evaluated 200 reasoning traces from Qwen3-14B (100 explicitly corrected, 100 consistently followed) across all intervention types on MMLU for Intra-Draft Faithfulness.
>
> **Human-Human Agreement Rates:**
> | Task | Aveg Agreement (%) |
> |---   | --- |
> | Step Labeling | 90.7 |
> | Behavior | 81.0 |
> | Answer Extraction | 99.3 |
>
> **Human-LLM Agreement (using Majority Vote from human annotations):**
> | Task | Agreement (%) |
> | --- | ------------- |
> | Step Labeling | 91.5 |
> | Behavior | 81.0 |
> | Answer Extraction | 100 |
>
> We also calculate **Cohen's Kappa coefficient** between the majority vote of human labels and the LLM for both step labeling and behavior classification. Step labeling yields a coefficient of 0.83, indicating **almost perfect agreement**. Behavior classification, which is more challenging for humans due to the extended content in the reasoning trace, still yields a coefficient of 0.62, reflecting **substantial agreement**.
>
> Furthermore, we assessed reliability across multiple LLM judges (Qwen2.5-32B-Instruct, Llama-3.2-70B-Instruct, Qwen3-32B) for behavior classification on three additional models: Deepseek-R1-Distill-1.5B (DS-1.5B), Skywork-OR1-7B (OR1-7B), and Qwen3-14B (faithfulness rate reported in Answer 5) over all generated traces on both MMLU and GPQA:
>
> **Cross Model Agreement Rates:**
> | Models | Avg Agreement (%) |
> | --- | --- |
> | DS-1.5B |  91.5 |
> | OR1-7B | 81.1 |
> | Qwen3-14B | 82.4 |
>
> These high agreement rates, **comparable to human-level consistency**, strongly support the reliability of our LLM-based evaluation method.
>
> In addition to evaluating the judge’s reliability, we will include more qualitative examples of LLM-generated corrupt options and misleading answers, and we will open-source the full evaluation code and data in a later revision. Unfortunately, the current rebuttal phase does not allow external links to share these examples. Notably, similar approaches—using LLMs to inject mistakes—have been widely adopted in prior work [2], further supporting the reliability of this method for content modification
>
> ---
> ### **Answer 2 (Weakness 2):** Complexity of Faithfulness Definition.
> We acknowledge that the definitions of faithfulness may initially appear complex. This complexity arises from the **nature of faithfulness** itself, which is not directly observable. As in prior work [1,2], we adopt a decomposition strategy to make this concept operational.
>
> Our decomposition into Intra-Draft Faithfulness and Draft-to-Answer Faithfulness captures distinct, practically important dimensions that are essential for **monitoring, interpretability, and controllability** in reasoning models (see Lines 49–64 in Section 1 and examples in Figure 1, Section 3).
>
> Specifically, **Intra-Draft Faithfulness** measures both local dependencies (e.g., explicit correction) and global dependencies (e.g., consistent following) among intermediate reasoning steps within the thinking draft. **Draft-Answer Consistency** assesses whether the model’s final answer aligns with the logical conclusion of the draft, while **Draft-Reliant** evaluates whether the answer stage depends entirely on the thinking draft.
>
> To further clarify these distinctions, we will expand our discussion of the motivations behind each dimension in the later revision.
>
> ---
>
> ### **Answer 3 (Weakness 3 & Question 1):** Incorporating Internal State Analysis
>
> Thank you for this insightful suggestion. To deepen our analysis, we now include experiments correlating internal model calculations (perplexities) with the observed selective integration phenomena of reasoning steps in Intra-Draft Faithfulness.
>
> Specifically, we compute the **conditioned perplexity difference** given a question $x$, an original thinking draft without counterfactual insertion $(T_{-}, T_{+})$, and  $T'\_{-}$, a variant of $T_{-}$ with counterfactual insertion:
>
> $$\Delta\text{PPL}(x, T\_{-}, T'\_{-},  T\_{+}) = \text{PPL}(T\_{+}  | x, T'\_{-}) - \text{PPL}(T\_{+}|x, T\_{-})$$
>
> Where $\text{PPL}$ represents the conditioned perplexity calculation over question and thinking draft. All results are computed based on the conditioned perplexity over the **first 100 tokens** of $T_{+}$ to ensure a fair comparison. We interpret a higher $\Delta\text{PPL}$ as indicating **greater integration** into the reasoning process, since the original continuation becomes less likely under the perturbed context introduced by the intervention.
>
> We run experiments on all test traces on MMLU and GPQA, and report the average $\Delta\text{PPL}$ for both continue and backtrack insertion types, comparing first/middle versus end insertion locations, using DS7B, OR1-7B, and Qwen3-14B.
>
>
> | Model | Continue End  | Continue First/Mid  | Backtrack End  | Backtrack First/Mid|
> |---|---|---|---|---|
> | OR1-7B | 1.30 | 0.71 | 1.86 | 0.89 |
> | DS-7B | 2.39 | 0.72 | 3.03 | 0.89 |
> | Qwen3-14B | 1.67 | 0.49 | 1.81 | 0.62 |
>
> These preliminary results show a significantly higher $\Delta\text{PPL}$ for end-step insertions compared to first/middle-step insertions, supporting our finding of location-dependent faithfulness in reasoning steps. Furthermore, the consistently higher $\Delta\text{PPL}$ for backtrack insertions corroborates our conclusion that backtracking steps are more faithfully integrated. We will incorporate a more detailed analysis of internal states and additional experiments into the revision to strengthen these findings.
>
> ---
>
> ### **Answer 4 (Question 2):** Propose Methods for Improving Faithfulness
> Thank you for appreciating our analysis! We agree that exploring concrete methods for improving faithfulness is critical. Based on our findings, promising future directions include:
>
> * Carefully design RLVR rewards directly tied to our defined faithfulness metrics.
>
> * Extending thinking interventions [4] to encourage sustained attention to reasoning steps, thereby enhancing intra-draft faithfulness.
>
> However, as demonstrated by prior research [3], effectively enhancing model faithfulness is inherently challenging and involves substantial additional complexity beyond our current scope. Nonetheless, our work offers foundational analyses and measurement tools that we believe are essential for guiding future advancements in this area.
>
> ---
>
> ### **Answer 5 (Question 3):**  Strengthening the Evaluation Benchmark
> Thank you for this valuable suggestion! To enhance our evaluation benchmark, we incorporated additional evaluations using the MMLU College Math subset (100 college math reasoning questions) and included new models (Qwen3-14B, OR1-7B, DS-1.5B).
>
> **MMLU College Math Results (%):**
> |   |Intra-Draft   |Draft-Reliant   |Draft-Answer   |
> |---|---|---|---|
> |  DS-1p5b | 48.0 |74.4   | 37.6  |
> |  OR1-7B | 49.2 | 76.0 |19.2   |
> |  DS-7B |55.8 |57.1   |36.9   |
> |  DS-8B| 53.4   |72.5   |29.2   |
> |  DS-14B|59.1  |84.6   |34.1   |
> |  Qwen3-14B| 55.3  |95.0   | 4.4  |
> |  DS-32B| 61.5  |85.6   |30.8   |
> |  QwQ-32B| 43.8  |95.5   |4.6   |
> |  OR1-32B | 56.3 |90.2 |14.8  |
>
> Additionally, we report the following results for GPQA and MMLU global facts subset for newly added tested models:
>
>  **GPQA Results (%):**
> |   |Intra-Draft   |Draft-Reliant   |Draft-Answer   |
> |---|---|---|---|
> |  DS-1.5b | 40.2 |71.0  | 42.7  |
> |  OR1-7B | 40.7 | 59.5 | 39.4  |
> |  Qwen3-14B| 44.2  |81.3   | 24.6  |
>
> **MMLU Global Facts Results (%):**
> |   |Intra-Draft   |Draft-Reliant   |Draft-Answer   |
> |---|---|---|---|
> |  DS-1.5b | 40.3 |76.3  | 48.1  |
> |  OR1-7B | 47.7 | 68.1 | 72.7  |
> |  Qwen3-14B| 58.5  |84.5   | 39.7  |
>
> These expanded evaluations consistently confirm our original observations. For instance, larger model in same model family general presents higher Intra-Draft Faithfulness, RLVR tuning incentive stronger internal preferences with low Draft-Answer Consistency, and math reasoning, which is more complex than factual recall over MMLU-global facts, result less faithful rate in Intra-Draft faithfulness and Draft-Answer Consistency. These observations of strengthened benchmarks reinforcing the reliability and generalizability of our conclusions. We will include more analysis over these experimental results within later revision.
>
> ---
> Thank you again for your constructive comments, which have helped us further improve our manuscript.
>
> [1] Chen et al. Reasoning Models Don't Always Say What They Think
>
> [2] Lanham et al. Measuring Faithfulness in Chain-of-Thought Reasoning
>
> [3] Tanneru et al. On the Hardness of Faithful Chain-of-Thought Reasoning in Large Language Models
>
> [4] Wu et al. Effectively Controlling Reasoning Models through Thinking Intervention

---

> > ### Author Response · Authors · 2025-08-05
> >
> > Dear Reviewer hiiz:
> >
> > Thank you again for your thoughtful and detailed review of our paper. We sincerely appreciate the time and effort you have devoted to evaluating our work.
> >
> > As the Author–Reviewer discussion phase nears its end, we would like to kindly ask whether our responses have addressed your concerns. We hope to have enough time to provide further clarification or conduct additional analyses if you have any remaining questions.
> >
> > In particular, to address your earlier points, we have added several new experiments and results:
> >
> > * A reliability analysis of using LLMs as evaluators for our proposed faithfulness metrics (**Answer 1**);
> > * New internal calculation measurements to support our claims (**Answer 3**).
> > * Construction of an additional evaluation subset and inclusion of three more models (**Answer 5**);
> >
> > Please let us know if these additions resolve your concerns or if there are any remaining issues you would like us to address.
> >
> > Best,
> >
> > Authors

---

> > ### Comment · Reviewer_hiiz · 2025-08-09
> >
> > Thank you for your feedback. I'm glad to see the new experimental results and I will revise the score upward to better align with these findings.

---

> > > ### Author Response · Authors · 2025-08-09
> > >
> > > Thank you for your response! We are glad that we were able to address your concern and sincerely appreciate your adjustment of the score!

---

### Note · Authors · 2025-08-11

We thank again the AC for handling our work and reviewers for their careful reading and constructive comments and discussion.

We are glad several reviews highlighted the value and strengths of our study, including:

* The importance of studying LRM faithfulness (Reviewers hiiz, VpmL, VqHF).
* The breadth and control of our evaluation and analysis (all reviewers).
* The use of diverse metrics to capture faithfulness properties (Reviewer VpmL).
* The intervention and experimental design (Reviewer VpmL, wfk8).
* Insights from counterfactual analysis, such as stronger faithfulness to backtracking steps (Reviewers VpmL, wfk8).
* Clear structure and presentation (Reviewers hiiz, VpmL).

Reviewers also raised concerns including LLM-as-judge reliability, internal-state analysis, dataset scope, and the naturalness of interventions. During the discussion we added the following evidence and clarifications:

1. **LLM-as-judge reliability.** We ran a human and cross-model study showing high agreement between human labels and the LLM judge, with Cohen’s kappa indicating almost-perfect (step labeling) and substantial (behavior) agreement (Answer 1 to Reviewer hiiz).

2. **Benchmark scope.** We added **MMLU College Math** and three more models (Qwen3-14B, OR1-7B, DS-1.5B). The results follow our original trends, strengthening confidence in our measurements (Answer 5 to Reviewer hiiz).

3. **Internal-state signal.** We introduced a conditioned perplexity measure after intervention, which supports location-dependent effects and greater integration of backtracking steps (Answer 3 to Reviewer hiiz).

4. **Naturalness of intervention.** Unperturbed tests are desirable, but they are considerably challenging under our definition of faithfulness metrics. Our interventions are designed to create clear global dependencies that can be reliably captured by the LLM judge. We also report comparisons to concurrent simulatability-based setups and explain why the conclusions can differ. (Answer 2 to Reviewer VpmL and Answer 6 to Reviewer VqHF).

Across the Author–Reviewer discussion, we are glad that these additions and clarifications addressed most of the reviewers’ concerns. In the later revision, we will integrate all of these discussions.

We again appreciate the helpful comments from the reviewers, which helped improve our work, and the effort from the AC in handling our submission.

---

### Decision · Program_Chairs · 2025-09-17

**Decision:**

Accept (poster)

**Comment:**

This paper presents a counterfactual intervention framework to evaluate the faithfulness of intermediate reasoning traces (“thinking drafts”) in LRMs. Faithfulness is defined along two axes: (1) Intra-Draft Faithfulness—whether individual reasoning steps causally affect subsequent steps and the draft’s conclusion, and (2) Draft-to-Answer Faithfulness—whether the final answer aligns with and depends on the draft. Using interventions across six LRMs and two datasets (GPQA, MMLU), the study finds that models demonstrate selective faithfulness: they integrate backtracking steps more than forward-progressing reasoning, and often produce final answers that diverge from draft conclusions due to additional reasoning at the answer stage. The results highlight both the promise and the limitations of reasoning drafts as faithful explanations of model behavior.

Strengths:
1. The decomposition into Intra-Draft Faithfulness and Draft-to-Answer Faithfulness provides a structured lens for analyzing draft reasoning.
2. The study evaluates six diverse LRMs, spanning model sizes, families, and training methods, on datasets of different difficulty. This breadth strengthens generalizability.
3. The selective faithfulness to backtracking steps and divergence between drafts and final answers are novel contributions that deepen understanding of LRM reasoning processes.

Weaknesses:
1. The framework heavily depends on GPT-4o-mini and Qwen for generating counterfactuals, extracting answers, and labeling behaviors. This introduces potential biases and reliability issues that are insufficiently validated (e.g., lack of human annotation or inter-annotator agreement).
2. The counterfactual intervention approach builds on existing faithfulness evaluation methods (e.g., CoT faithfulness, simulability). The paper’s main novelty lies in applying these to reasoning drafts, which may limit impact.
3. Core functions and metrics (e.g., ϕ for interventions, ANS for extracting conclusions) are under-specified or confusingly decomposed. This reduces accessibility and reproducibility.

The paper tackles an important and under-explored question: whether LRM thinking drafts faithfully reflect the reasoning process leading to answers. Its structured framework, broad model coverage, and insightful findings make it a valuable contribution.

After the discussions, all the reviewers are positive on this work.